# KNN-based frequency-adjustable ferroelectric heterojunction and biomedical applications

Tao Zhang[1,6], Haoyuan Hu[2,3,6], Hong Jiang [2,3] ✉, Zhen Wang[4], Jinfeng Lin[5], Ye Cheng[2,3], Wei Guo[2,3], Di Ke[1], Hai Hang[1], Mengshu Ta[1], Jun Ou-Yang[1], Jiwei Zhai [5], Xiaofei Yang[1], Songyun Wang [2,3] ✉ & Benpeng Zhu [1] ✉

High-performance lead-free $K_{0.5}Na_{0.5}NbO_3$ piezoelectric ceramics present a practical alternative to lead-containing counterparts by effectively reducing potential environmental hazards. This advancement is particularly relevant to the development of ferroelectric heterojunction devices for biomedical applications. Here, we design and fabricate a frequency-adjustable ferroelectric heterojunction based on the developed $K_{0.5}Na_{0.5}NbO_3$ piezoelectric ceramics with a high piezoelectric coefficient ($d_{33} = 680$ pC/N). By leveraging flexible encapsulation, the heterojunction achieves miniaturization ($\varphi = 13.3$ mm, $h = 2.28$ mm) and suitability for implantation. After penetrating the rat skull, the ultrasound generated by the heterojunction at a frequency of 3 MHz reaches a focal depth of about 7.9 mm, a focal width of approximately 480 μm at −6 dB, and millimeter-scale continuous focal tuning (1.5 mm) within a narrow frequency range (2.7–3.3 MHz). Additionally, the implanted heterojunction enables long-term and high-precision transcranial neuromodulation, and consequently yields therapeutic effects in a myocardial infarction animal model. Collectively, this study highlights a viable strategy for developing and applying lead-free ferroelectric heterojunctions, expanding their potential in brain modulation, and providing new insights into clinical treatments of myocardial infarction.

Brain modulation is an effective strategy for elucidating the mechanisms of brain function, consequently, treating related disorders[1,2]. Compared with traditional techniques, such as optogenetics, transcranial magnetic stimulation, and transcranial electrical stimulation[3–5], transcranial focused ultrasound stimulation possesses unique advantages, including high penetration depth, high spatial resolution, and excellent biocompatibility[6–8]. As numerous neurological and cardiovascular disorders, such as Parkinson's disease, epilepsy, Alzheimer's disease, and myocardial infarction[9–11], exhibit chronic and

refractory characteristics, long-term, high-precision neuromodulation is required to ameliorate abnormal neural activity. Unfortunately, traditional transcranial focused ultrasound devices encounter substantial limitations in practical applications. Firstly, their large size and insufficient focusing precision render them unsuitable for fullimplantation[12,13]. Additionally, the fixed focal depth of existing devices[14–16] constrains precise targeting of specific brain regions due to discrepancies between device design and practical applications. Consequently, current transcranial focused

[1]School of Integrated Circuit, Huazhong University of Science and Technology, Wuhan, China. [2]Cardiovascular Hospital, Renmin Hospital of Wuhan University, Wuhan, China. [3]Cardiac Autonomic Nervous System Research Center, Wuhan University, Wuhan, China. [4]National Institute of Dental and Craniofacial Research (NIDCR), National Institutes of Health (NIH), Bethesda, MD, USA. [5]School of Materials Science and Engineering, Tongji University, Shanghai, China. [6]These authors contributed equally: Tao Zhang, Haoyuan Hu. ✉e-mail: hong-jiang@whu.edu.cn; wsy7982@126.com; benpengzhu@hust.edu.cn

ultrasound devices remain inadequate for long-term, high-precision neuromodulation.

To address these challenges, it is essential to develop a miniature, implantable transcranial ultrasound device with adjustable focus. Recently, approaches for focused ultrasound have been proposed, primarily involving active and passive wavefront manipulation techniques. In active wavefront manipulation, since ultrasound phased arrays play an important role, where complex electrical system should be included[17–19], they are not conducive for implantable applications. Although the piezoelectric reverse polarization method is likewise an effective approach for active wavefront controlled focused ultrasound[20], impediments persist in the polarization process for miniaturized device. Regarding passive wavefront manipulation, acoustic lenses or acoustic metasurfaces are utilized to focused ultrasound[21–24], requiring only an electrically connected planar acoustic transducer. Compared to acoustic lenses, acoustic metasurfaces are increasingly adopted because of the advantages of subwavelength planar structures[25–27]. Furthermore, the design of active spiral Fresnel zone plates provides opportunities for adjustable-focus ultrasonic devices[28]. Therefore, acoustic metasurfaces based on the Fresnel spiral principle present a promising solution for achieving both miniaturization and adjustable-focus in transcranial ultrasound devices. Until now, however, associated research remains limited.

Additionally, piezoelectric materials are the core of ultrasonic devices, which is important for the development of ultrasonic heterojunction. Unfortunately, current piezoelectric devices mainly rely on lead-based piezoelectric materials [e.g., Pb (Zr, Ti) O$_3$, PZT] as acoustic-electric coupling elements[29], raising concerns over potential health hazards. In recent years, a growing number of studies have concentrated on lead-free piezoelectric ceramics, with potassium sodium niobate (K$_{0.5}$Na$_{0.5}$NbO$_3$, KNN)- based piezoelectric ceramics[30–33] emerging as one of the most promising alternatives to lead-based ones. Inspired by the morphotropic phase boundary in PZT ceramics, the construction of polymorphic phase boundary by chemical modification can significantly improve the piezoelectricity[34]. Moreover, the entropy increase strategy[35,36], in which multiple elements occupying equivalent lattice positions has recently provided key perspectives on the design of high-performance piezoelectric ceramics.

In this study, we conceive and fabricate a frequency-adjustable ferroelectric heterojunction (f-FH) for transcranial neuromodulation (Fig. 1a), which achieves continuous adjustable-focus at different operating frequencies (Fig. 1b). Furthermore, the f-FH is based on the developed KNN piezoelectric ceramics with a high $d_{33}$ of 680 pC/N and the structural design of a Fresnel spiral diffraction grating (Fig. 1c), reaching continuous focus adjustment in the millimeter range (1.5 mm) within a narrow frequency range (2.7–3.3 MHz). Additionally, the f-FH is miniaturized ($\varphi = 13.3$ mm, $h = 2.28$ mm) (Fig. 1d), allowing implantation in animals for long-term and high-precision transcranial neuromodulation targeting the paraventricular nucleus of the

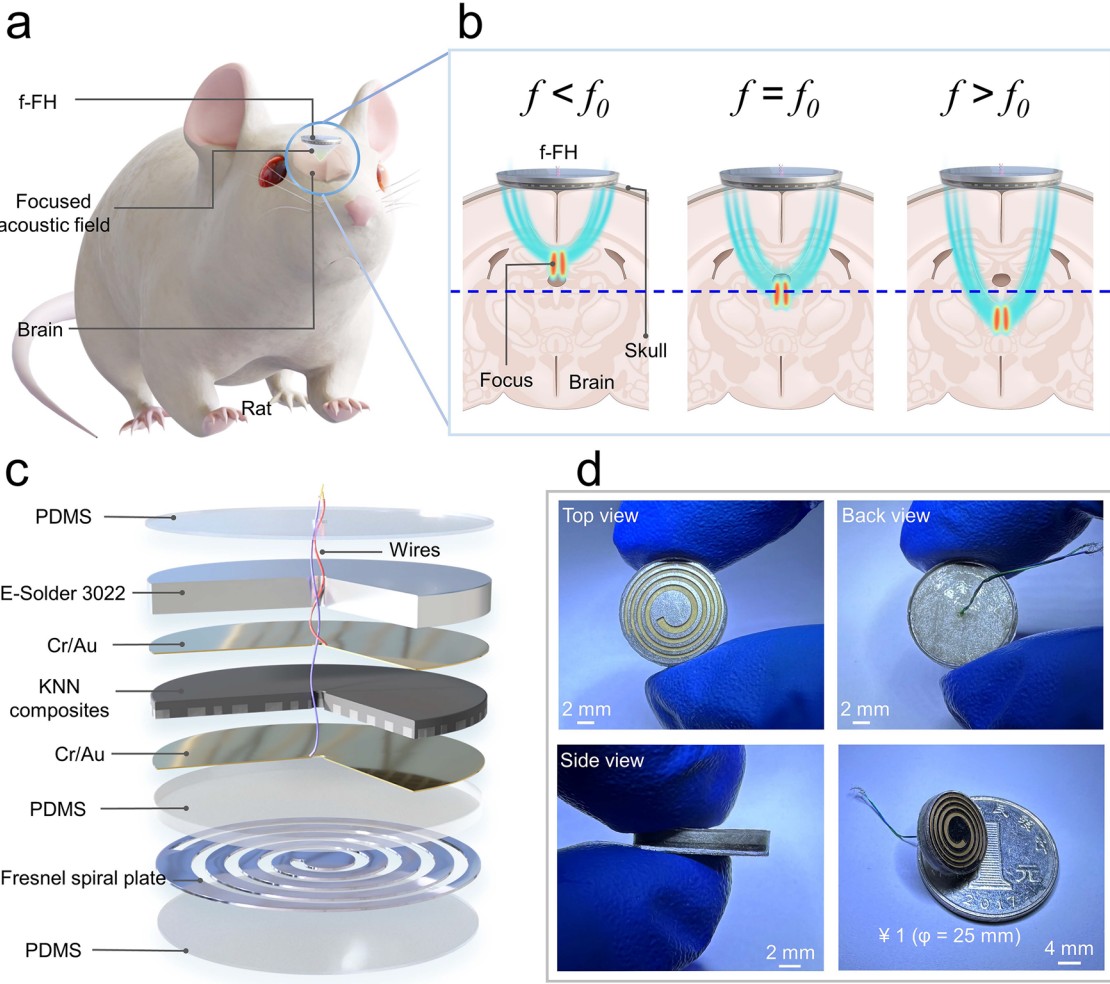

**Fig. 1 | Principle and application of the frequency-adjustable ferroelectric heterojunction (f-FH). a** Diagram of neuromodulation by f-FH. **b** Schematic diagram of f-FH's operating frequency to adjust the focus depth. **c** Schematic diagram of the structure of f-FH. **d** Top view, back view, side view of the f-FH and the miniaturized f-FH standing on a one-yuan coin.

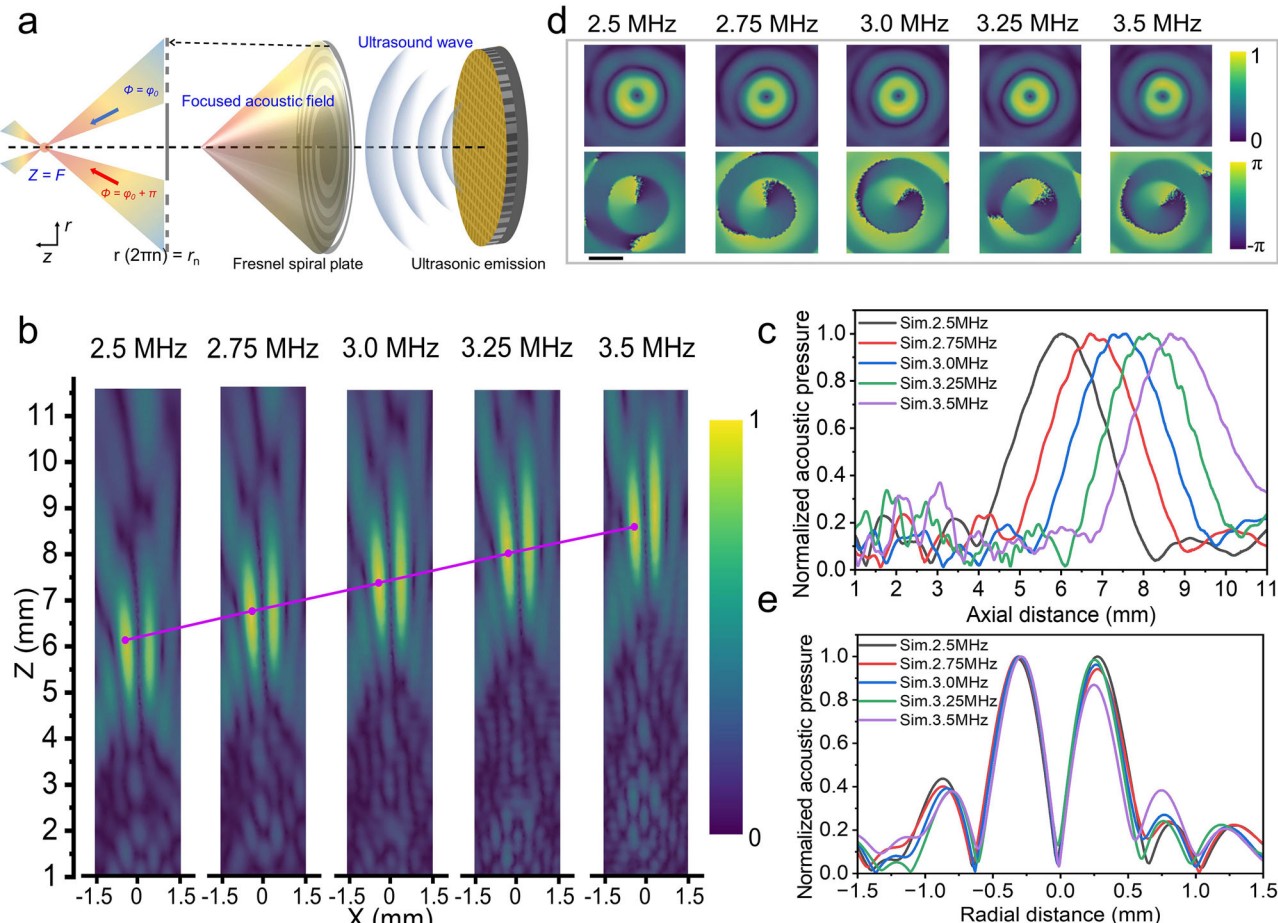

**Fig. 2 | Theory and simulation of the f-FH. a** The Fresnel spiral diffraction grating structure of the f-FH. **b** Simulations results for the acoustic pressure on the $y = 0$ plane radiated by the f-FH, at $f = 2.5$, 2.75, 3.0, 3.25 and 3.5 MHz. **c** Normalized axial pressure distribution of simulations. **d** Simulations results for the acoustic pressure and phase on the focus plane radiated by the f-FH, at $f = 2.5$, 2.75, 3.0, 3.25 and 3.5 MHz. **e** Normalized pressure distribution at the focal plane for the simulation.

hypothalamus (PVN), enabling therapeutic benefits for MI. Collectively, we develop a lead-free ferroelectric heterojunction with expanded potential in brain modulation, providing new insights into clinical treatments of MI.

## Results and discussions
### Design and field simulation of the f-FH
To address the key limitations of conventional transcranial focused ultrasound devices in neuromodulation applications, we propose a frequency-adjustable ferroelectric heterojunction (f-FH) for transcranial neuromodulation, which achieves continuous adjustable-focus at different operating frequencies (Fig. 1b). By integrating ferroelectric KNN with a Fresnel spiral diffraction grating (Fig. 1c), the device achieves a compact, fully implantable form factor.

We employ an acoustic metasurface to attain miniaturization, as described previously[25–27]. Specifically, the metasurface adopts a Fresnel-spiral diffraction grating structure to generate a focused acoustic field (Fig. 2a). According to Fresnel-spiral diffraction theory, the boundary of the Mth arm is defined by the following equation[37]:

$$r_1(\theta)^2 = \left[\sqrt{r_0^2 + F_0^2} + \left(\frac{M\theta}{2\pi} - m\right)\lambda_0\right]^2 - F_0^2 \tag{1}$$

$$r_2(\theta)^2 = \left[\sqrt{r_0^2 + F_0^2} + \left(\frac{M(\theta+\pi)}{2\pi} - m\right)\lambda_0\right]^2 - F_0^2 \tag{2}$$

Where $F_0$ is the focal length; $\lambda_0$ is the wavelength; $M$ (set as 1 in this experiment, $M = 1$, $m = 1$) is the topological charge of the sound vortex; $r_0$ is the radius of the central opaque region ($r_0^2 = (F_0 + \lambda_0)^2 - F_0^2$). With this Fresnel spiral grating, vortex focusing is reached based on phase dislocation.

For the ultrasound source, high-frequency ultrasound exhibits a sharp increase in attenuation following skull penetration[38]. In animal experiments, transcranial ultrasound typically operates at various frequencies ranging from 500 kHz to 4 MHz[39–43]. As derived from the theoretical framework above, when the focal depth remains constant, lower-frequency ultrasound yields a longer wavelength, thereby increasing the Fresnel spiral diameter. Given that miniaturization is a critical requirement for implantable devices (focal length: 7.8-8.0 mm, PVN brain area)[44], the ultrasound frequency ($f_0$) is set to 3 MHz. Moreover, when the device operates at an alternative frequency ($f$), the focal length ($F$) shifts as described by the following equation[28]:

$$F(f) = \alpha F_0 + \frac{M\lambda_0}{2}\left(\alpha - \frac{1}{\alpha}\right) \tag{3}$$

Where $\alpha = f/f_0$. Hence, the focal length can be tuned by adjusting the excitation ultrasound frequency.

Figure 2b displays the simulated pressure field amplitude of the f-FH on the $y = 0$ plane. As the frequency increases, the focal length varies nearly linearly, extending to 7.4 mm at $f = 3$ MHz (Fig. 2c). To characterize the vortex properties of the focused acoustic beam, we

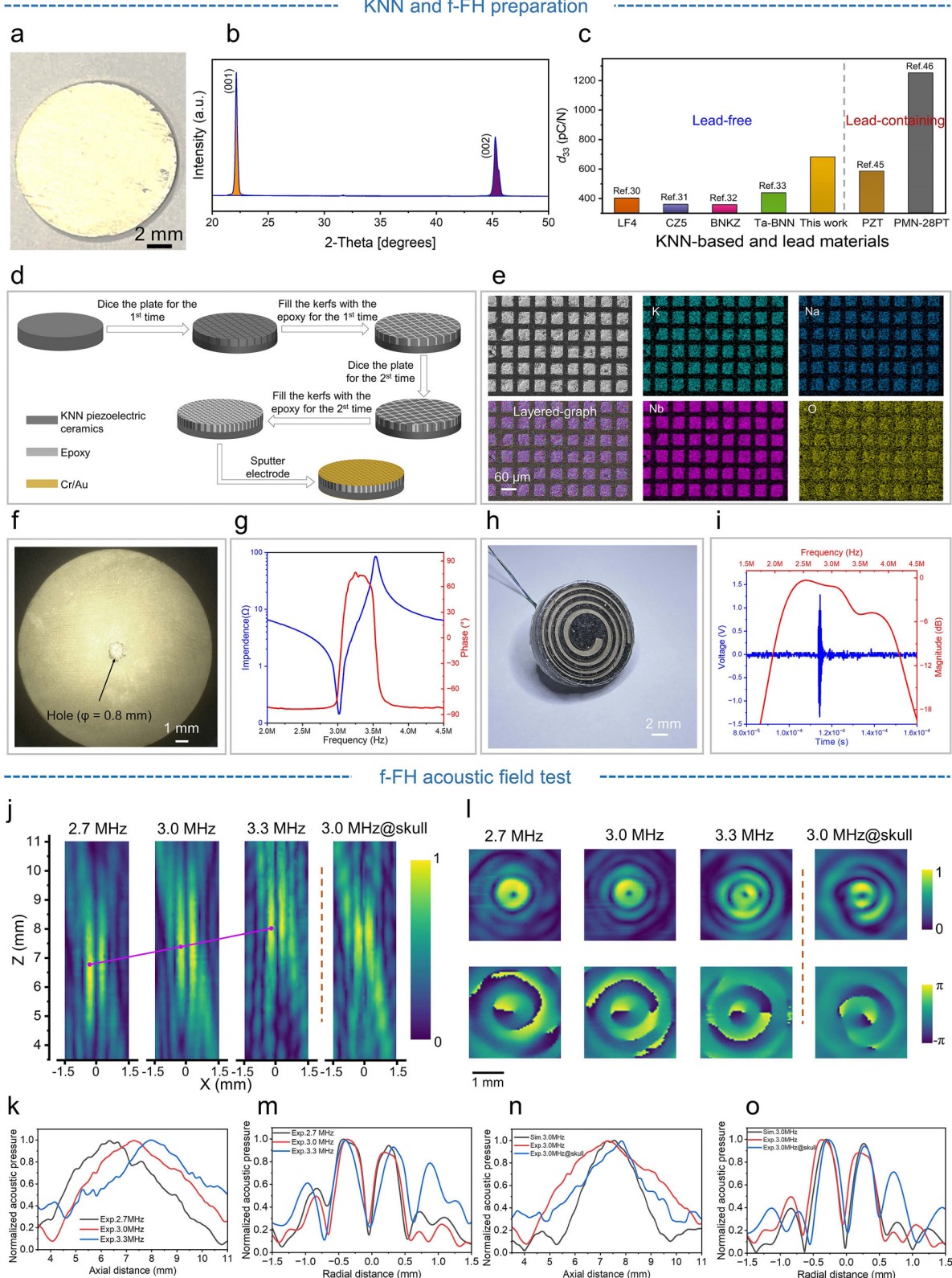

Fig. legend region with labels: KNN and f-FH preparation; f-FH acoustic field test

analyze the amplitude and phase distributions of the pressure field on the focal plane (Fig. 2d, e). The pressure amplitude is minimal at the center, surrounded by a distinct phase vortex.

## Preparation and examination of KNN-based f-FH

As previously mentioned, the core component of the f-FH is a KNN-based ferroelectric ceramic. For deep brain neuromodulation, KNN

piezoceramics with high piezoelectricity are required to ensure effective acoustic pressure output. Thus, based on the entropy phase transition strategy, ferroelectric KNN ceramics with local polymorphic distortion of new phase boundary (NPB) are designed. Specifically, starting from the orthorhombic perovskite-type $K_{0.5}Na_{0.5}NbO_3$ matrix, medium-entropy piezoelectric ceramics (Fig. 3a) are developed by introducing Sb, Zr, Hf and Ti at the B-site and Ca and Bi at the A-site

**Fig. 3 | Design and fabrication of the KNN piezoelectric ceramic and f-FH.**
**a** Developed KNN-3.5BHT piezoelectric ceramics. **b** XRD patterns of the KNN-3.5BHTceramic. **c** Piezoelectric performance comparison of the KNN-3.5BHT ceramics with other piezoelectric ceramics. **d** Preparation of KNN piezoelectric composites. **e** SEM images and corresponding elemental mapping of KNN composites. **f** Image of the sputtered Cr/Au electrodes and the piezoelectric composite after perforation. **g** Electrical impedance and phase angle of the KNN piezoelectric composite. **h** Photo of the f-FH. **i** The echo signal and frequency gain of the f-FH. **j** Experimental results for the acoustic pressure on the $y = 0$ plane radiated by the f-FH, at $f = 2.7$, 3.0, 3.3 MHz without skull and 3.0 MHz with skull. **k** Normalized axial pressure distribution of experiments. **l** Experimental results for the acoustic pressure and phase on the focus plane radiated by the f-FH, at $f = 2.7$, 3.0, 3.3 and 3.0 MHz with skull. **m** Normalized pressure distribution at the focal plane for experiments. **n** Comparison of normalized axial pressure distribution on the path through the focus at $f = 3.0$ MHz between simulations and experiments. **o** Comparison of normalized axial pressure distribution at the focal plane between simulations and experiments.

(Supplementary Fig. 1a), combined with the application of templated grain growth technology. The resulting composition of the as-prepared material is $(K_{0.505}Na_{0.5-95.5\%} Ca_{0.01}Bi_{0.5-3.5\%}) (Nb_{0.965-95.5\%} Sb_{0.035-95.5\%}Zr_{0.01}Hf_{0.98-3.5\%}Ti_{0.02-3.5\%}) O_3$ (abbreviated as KNN-3.5BHT). Crucially, the KNN-3.5BHT ceramics demonstrate excellent compositional homogeneity (Supplementary Fig. 1b, c).

The XRD results (Fig. 3b) indicate that the KNN ceramic adopts a perovskite structure, characterized by prominent <001> and <002> peak intensities. Generally, the incorporation of Bi, Hf and Ti into the lattice elevates the configurational entropy of the ceramic system, which drives phase transitions. The evolution of the local structure is further corroborated by Raman spectroscopy. In the KNN ceramic, the observed Raman scattering peaks predominantly arise from $NbO_6$ octahedral vibrations. Compared to pure KNN, the relative intensity of scattering peaks (marked by arrows) decreases gradually in the high-entropy KNN-3.5BHT, suggesting elevated local structural disorder and relaxation effects due to entropy enhancement (Supplementary Fig. 2a). Notably, the piezoelectric constant ($d_{33}$) of pure KNN is only 150 pC/N, whereas KNN-3.5BHT reaches a significantly enhanced $d_{33}$ of 680 pC/N (Supplementary Fig. 2b). In contrast to other KNN-based lead-free piezoelectrics[30–33], the KNN-3.5BHT ceramics demonstrate superior performance (Fig. 3c). Remarkably, even in comparison to lead-based counterparts[45,46], the piezoelectric properties of KNN-3.5BHT surpass those of widely used PZT, establishing a strong foundation for the development of high-performance lead-free ferro-electric heterojunctions.

To facilitate optimal device miniaturization, we employ a piezo-electric composite material with lower acoustic impedance [47–49]. This design minimizes sound reflection at the interface with the low-impedance biocompatible polymethylsilsesquioxane (PDMS) encapsulation layer[50]. As shown in Fig. 3d, e, we fabricate piezoelectric composites using the developed ferroelectric KNN (Supplementary Fig. 3, and Table 1). Following cutting and electrode sputtering, the resulting composites have a diameter of 13 mm (Fig. 3f, and Supplementary Fig. 3), and demonstrate a resonance peak frequency of 3 MHz (Fig. 3g). The final flexible f-FH is constructed by integrating the acoustic components with flexible PDMS packaging (Supplementary Fig. 4), yielding a compact form factor (Fig. 3h). The complete assembly weighs merely 1.0 g, with dimensions of 13.3 mm in diameter and 2.28 mm in thickness (Supplementary Fig. 5). Acoustic characterization reveals that the f-FH operates at a central frequency of 3 MHz with an impressive 60.5% bandwidth (Fig. 3i), aligning with the experimental requirements.

Figure 3j displays the experimentally measured pressure field amplitude of the f-FH on the $y = 0$ plane. In the absence of the skull, the focal length rises approximately linearly with frequency (Fig. 3k), and the amplitude and phase distributions on the focal plane (Fig. 3l, m) closely match the simulation results. Although the presence of skull distorts the acoustic field distribution, the system maintained robust focusing performance, with the focal length increasing by nearly 0.5 mm (Fig. 3n). Moreover, after penetrating the rat skull, the focusing beam attains continuous millimeter-scale (1.5 mm) focal adjustment within a narrow frequency band (2.7–3.3 MHz) (Supplementary Fig. 6), and the focal width at 3.0 MHz reaches 480 μm at −6 dB (Fig. 3o). Furthermore, the excitation voltage and transcranial focal pressure

show an approximately linear relationship (Supplementary Fig. 7), with an acoustic pressure of 0.36 MPa at 100 V, enabling applications in deep brain neuromodulation.

## MI treatment applications of the f-FH

Myocardial infarction (MI) remains a leading cause of global mortality[51], as post-MI adverse cardiac remodeling ultimately leads to cardiac dysfunction and heart failure[52,53]. Previous studies suggest that the PVN stimulation may hold therapeutic potential for MI[54]. Here, utilizing the f-FH, we apply long-term transcranial ultrasound modulation to the bilateral PVN of rats with MI (Fig. 4a).

Rats are randomized into three groups: the control group ($n = 12$, sham device implantation and sham MI), MI group ($n = 12$, sham device implantation and MI), and MI + f-FH group ($n = 12$, device implantation and MI). Following f-FH fixation (Fig. 4b), rats undergo a recovery for two weeks before MI induction via ligation of the left anterior descending coronary artery. The MI + f-FH group subsequently receives daily 15-minute transcranial ultrasound stimulation of the PVN for four consecutive weeks (Supplementary Movie 1, Supplementary Fig. 8). Given that low-intensity focused ultrasound alleviates PVN neuroinflammation[54,55], the ultrasound stimulation intensity for MI treatment ($I_{MI}$) by f-FH is set to 270.6 mW/cm² (Supplementary Table 2). Additionally, rats in all groups receive electrophysiological and histological assessments, and PVN slices are collected for further analysis (Supplementary Fig. 9).

Neuronal activity in the left stellate ganglion (LSG) is recorded (Fig. 4c, d). As sympathetic nerves regulate cardiac physiological activity, the LSG is overactive due to chronic MI, manifested by increased discharge amplitude and frequency. After long-term ultra-sound intervention on the PVN, inhibition of MI-induced excessive LSG activation is observed ($p < 0.01$) (Fig. 4e–g). Since the level of tyrosine hydroxylase (TH) in the PVN reflects the sympathetic neuronal activity[56], TH immunofluorescence staining is performed in the PVN across the three groups (Fig. 4h). Compared to the control group, MI leads to a significant elevation of TH+ neurons in the PVN ($p < 0.001$), which is suppressed by long-term f-FH stimulation ($p < 0.01$) (Fig. 4i). Furthermore, microglia and astrocytes, the immune cells in the central nervous system (CNS), are involved in neuroinflammatory responses in the PVN[57–59]. The purinergic ligand-gated ion channel 7 receptor (P2X7R) expressed in glial cells is closely associated with neuroinflammation[60,61]. To further investigate the effect of ultrasound stimulation on neuroinflammation, we perform immunofluorescence double-staining for P2X7R and Iba-1/GFAP to detect the expression of microglia/astrocytes and their surface P2X7R in the PVN (Supplementary Fig. 10, and Supplementary Fig. 11). The results indicate that microglia and astrocytes are significantly activated ($p < 0.001$) and the expression of P2X7R is upregulated after MI. Hence, following long-term ultrasound modulation by the f-FH, the inflammatory response in the PVN is suppressed.

To examine the effect of ultrasound intervention on sympathetic neural function, we measure serum levels of norepinephrine (NE), neuropeptide Y (NPY), and heart rate variability (HRV). As important serum markers of the sympathetic nervous system[62], the expression levels of NE and NPY are significantly higher in the MI group than in the control group ($p < 0.001$), while long-term ultrasound intervention

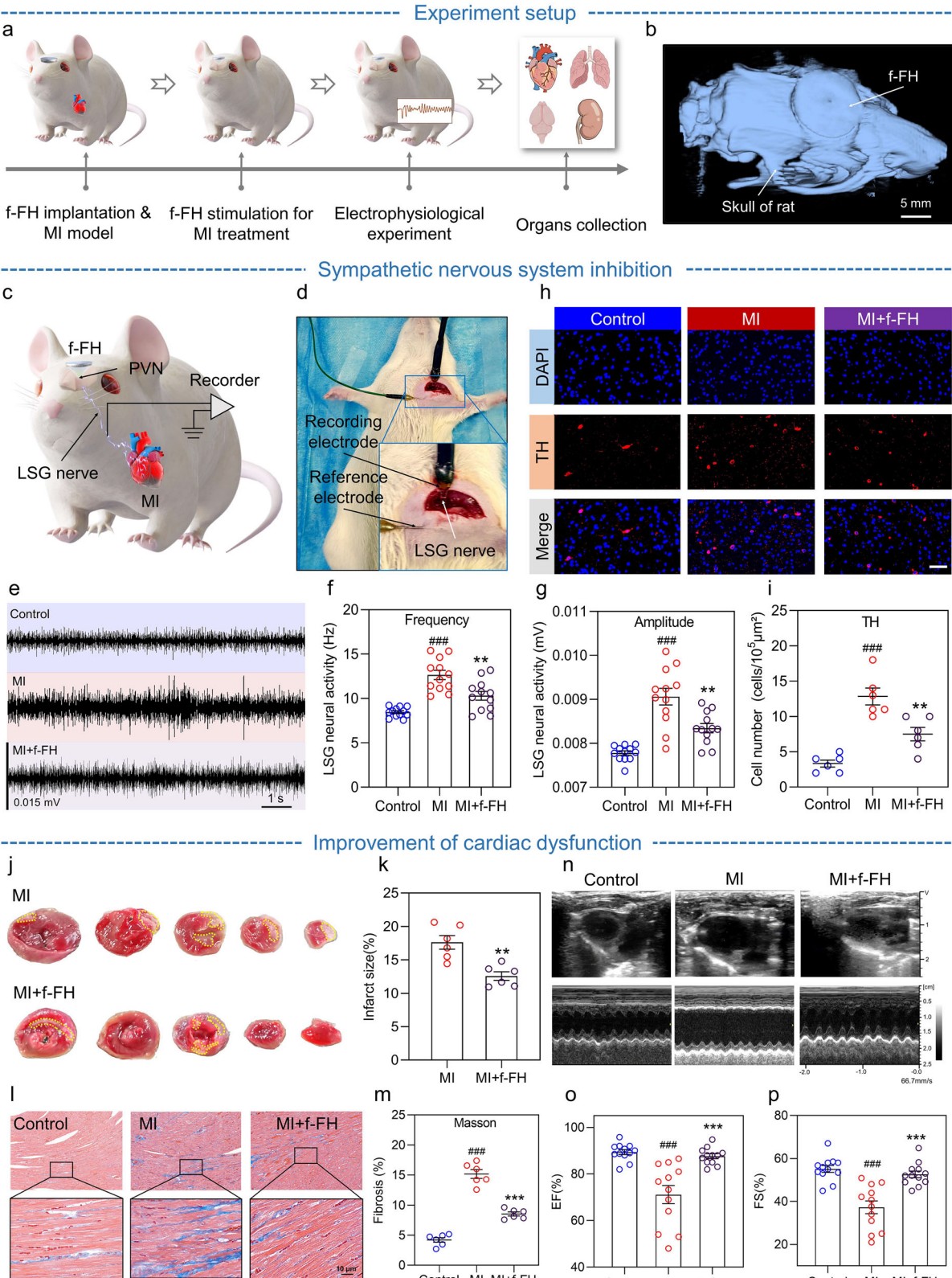

**Fig. 4 | MI treatment applications of the f-FH. a** Experimental flowchart. **b** CT three-dimensional image of the f-FH implanted in vivo. **c** Schematic illustration of LSG neural activity recording. **d** The animal experiment of LSG neural activity. **e** Typical schematic of LSG neural activity recording. **f, g** Statistical analysis of the frequency/amplitude of LSG neural activity ($n$ = 12). **h** Representative images of TH immunofluorescence staining in the PVN, scale bar = 50 μm. **i** Quantitative analysis of TH+ neurons in the PVN ($n$ = 6). **j** Representative images of myocardial TTC staining. The white areas (marked by yellow dashed lines) represent infarct myocardium. **k** Quantitative statistical analysis of myocardial infarct size ($n$ = 6). **l** Schematic representation of myocardial Masson's staining. The blue areas represent fibrous tissue. **m** Statistical analysis of myocardial fibrosis degree ($n$ = 6). **n** Representative images of B-mode and M-mode echocardiography. **o, p** Statistical analysis of EF% (ejection fraction), and FS% (fraction shorting) ($n$ = 12). Compared to the control group, $^{###}p < 0.001$; compared to the MI group, $^{**}p < 0.01$, $^{***}p < 0.001$.

reduces serum levels of NE ($p < 0.01$) and NPY ($p < 0.001$) (Supplementary Fig. 12). HRV, a key indicator of autonomic nervous tone, includes high frequency (HF), low frequency (LF), and the ratio of LF to HF (LF/HF). Compared to the control group, MI significantly enhances LF ($p < 0.001$) and LF/HF ($p < 0.001$) but decreases HF ($p < 0.01$). Conversely, ultrasound modulation dampens these changes induced by MI (LF: $p < 0.001$; HF: $p < 0.01$; LF/HF: $p < 0.001$) (Supplementary Fig. 13). All these findings illustrate that long-term transcranial ultrasound modulation on the PVN by f-FH improves sympathetic neural activity and function.

We further explore the effects of ultrasound intervention on ventricular structure, electrophysiological remodeling, and left ventricular dysfunction. Firstly, we assess the infarct size using triphenyltetrazolium chloride (TTC) staining in both the MI group and MI + f-FH group. Typical TTC staining images are shown in Fig. 4j. Compared to the MI group (17.62% ± 2.51%), ultrasound stimulation diminishes myocardial infarct size in the MI + f-FH group (12.57% ± 1.57%, $p < 0.01$) (Fig. 4k). Additionally, transmission electron microscopy (TEM) is used to evaluate the ultrastructure of myocardial tissue. Following MI, myocardial fiber rupture, mitochondrial swelling, and cellular matrix edema are observed, while these structural damages to myocardium are ameliorated using ultrasound modulation on the PVN (Supplementary Fig. 14). Excessive fibrosis after MI causes adverse ventricular remodeling, contributing to the development of heart failure[63]. Masson's trichrome staining is also applied to assess interstitial fibrosis in the infarct border zone of the left ventricle (LV) (Fig. 4l). MI significantly elevates the level of myocardial interstitial fibrosis ($p < 0.001$), while ultrasound stimulation mitigates the extent of fibrosis ($p < 0.001$) (Fig. 4m). Following myocardial infarction, impaired cardiac blood supply incurs irreversible myocardial cell damage. Therefore, increasing myocardial vascular density plays a crucial role in restoring myocardial blood supply and nutrient delivery[64,65]. Smooth muscle actin (α-SMA) and von Willebrand factor (vWF) are used to collectively label neovascularization within the myocardium, as shown in typical immunofluorescence double-staining images in Supplementary Fig. 15a. Quantitative analysis shows that compared to the MI group, f-FH intervention increases α-SMA+ ($p < 0.05$) and vWF+ vascular density ($p < 0.01$) (Supplementary Figs. 15b, c). These results indicate that long-term ultrasound modulation by the f-FH reverses ventricular structural remodeling after MI.

Furthermore, programmed electrophysiological stimulation (PES) is used to assess ventricular arrhythmia (VA) inducibility. In detail, programmed continuous pacing does not induce significant arrhythmias in the control group, while sustained ventricular tachycardia (VT) occurs in the MI group. After ultrasound stimulation, PES only incurs ventricular premature beats (VPB) (Supplementary Fig. 16). These results suggest that long-term ultrasound intervention on the PVN lowers susceptibility to VA after MI, thereby improving ventricular electrophysiological stability and inhibiting ventricular electrophysiological remodeling. Echocardiography is also performed to evaluate the effect of ultrasound intervention on cardiac function following MI. Representative B-mode and M-mode ultrasound images are presented in Fig. 4n. Compared to the control group, MI leads to a decrease in EF% and FS% (Fig. 4o, p), as well as an increase in LVESV, LVEDV, LVIDs, and LVIDd (Supplementary Fig. 17). MI causes left ventricular enlargement, thinning of the ventricular wall and deterioration of cardiac function, which is mitigated by ultrasound stimulation, consistent with macroscopic heart observation (Supplementary Fig. 18).

The above findings show that transcranial ultrasound neuromodulation of the PVN using the f-FH can effectively facilitate the treatment of MI. Furthermore, since the ultrasound intensity (270.6 mW/cm²) and mechanical index (0.2) delivered by the f-FH are both less than the FDA's safety threshold (720 mW/cm², and 1.9)[66], and the use of an interval pulse ultrasound stimulation, no significant

temperature changes occur during transcranial ultrasound stimulation (Supplementary Fig. 19). Moreover, biocompatibility assessments indicate no notable adverse reactions (Supplementary Fig. 20, and Supplementary Movie 2) and surrounding tissue lesions (Supplementary Fig. 21) following the long-term implantation of f-FH. Histopathological examination of major organs after f-FH implantation shows no evidence of toxicity (Supplementary Fig. 22), and blood routine as well as liver and kidney function tests demonstrate no significant abnormalities (Supplementary Fig. 23, and Supplementary Fig. 24). These results, therefore, indicate that our device possesses considerable biocompatibility.

## Methods

### Acoustic field simulation of the f-FH

The acoustic field generated by f-FH is simulated using COMSOL Multiphysics V6.3. In the constructed Fresnel spiral grating, the parameter M is 1, the design frequency (*f*) is 3 MHz, the focal length (*F*) is 7.4 mm, and the source aperture is 13 mm. The grating is composed of stainless steel with a thickness of 100 micrometers. The Fresnel spiral lines of the grating satisfy the above formulas (1) and (2), maintaining a spiral angle of 8π. A normal displacement plane is set as the sound source, positioned one wavelength away from the grating plane. Water is modeled as the acoustic domain, with radiation boundary conditions applied to the external boundaries. The speed of sound in water is defined as 1500 m/s, and the density to 1000 kg/m³. In addition, the focal length variation characteristics of the model are explored by setting the excitation ultrasound frequencies to 2.5, 2.75, 3.0, 3.25 and 3.5 MHz.

### Preparation and test of the KNN

The ($K_{0.505}Na_{0.5(0.99-x\%)}$ $Ca_{0.01}Bi_{0.5\cdot x\%}$) ($Nb_{0.965(0.99-x\%)}$ $Sb_{0.035(0.99-x\%)}$ $Zr_{0.01}Hf_{0.98\cdot x\%}Ti_{0.02\cdot x\%}$) $O_3$ ($x = 0$, 3.5, abbreviated as KNN-$x$BHT) piezoceramics are designed and fabricated using the conventional solid-state method. All raw materials, including $K_2CO_3$ (Aladdin, 99.5%), $Na_2CO_3$ (Aladdin, 99.8%), $Nb_2O_5$ (Aladdin, 99.98%), $CaCO_3$ (Sinopharm, 99.99%), $Bi_2O_3$ (Alfa Aesar, 99.975%), $HfO_2$ (Aladdin, 99.99%), $ZrO_2$ (Aladdin, 99.99%), $TiO_2$ (Sinopharm, 99.8%), and $Sb_2O_3$ (Alfa Aesar, 99.9%) are ball-milled with ethyl alcohol and $ZrO_2$ balls after weighting. The ceramic slurry is prepared by mixing the dried calcined powder with ethanol/toluene co-solvents, organic binders, and 3 wt% high-quality NN templates. After 8 h of roller milling, the homogeneous slurry is casted using casting machine. The dried tapes are then cut, laminated, and pressed into pellets at 200–300 MPa and 60 °C for 10 min. Finally, all the pellets are undergone binder burnout at 600 °C (1 °C/min) followed by two-step sintering at 1180–1210 °C for 6–8 h in air. Given the advantages of AC polarization[67,68], the ceramic is first poled at 30 kV/cm by AC electric field at room temperature, and then the piezoelectric coefficient ($d_{33}$) is tested at room temperature using a quasi-static $d_{33}$ meter (ZJ-6A, Institute of Acoustics, China). The temperature fatigue resistance of piezoelectricity is evaluated using the GDPT-900A Variable Temperature Piezoelectric $d_{33}$ Measurement System (JKZC, China), and the elemental composition and distribution are investigated using an EPMA-8050G electron probe microanalyzer (EPMA) (JEOL, Japan).

### Manufacture and test of the f-FH

KNN composite materials are prepared using cutting and filling techniques with a DAD323 dicing saw (DISCO, Saitama, Japan) equipped with a 20-μm and blade and a cutting depth of 350 μm. Cr/Au (50/100 nm) electrodes are deposited on both sides of the polished composite material using sputtering techniques, and polarization is performed using a high-voltage DC power supply. Piezoelectric rings with an outer diameter of 13 mm and an inner diameter of 0.8 mm are cut through laser cutting and drilling. Flexible wires with a diameter of 300 μm are adopted for electrode interconnections, and the central hole is filled

with Epoxy 301 (Epoxy Technology, Billerica, MA). E-Solder 3022 (Von Roll Isola, New Haven, CT, USA) is served as the backing material with a thickness of 0.9 mm.

To generate an effective vortex sound field, the distance from the piezoelectric elements to the spiral diffraction grating is set to one wavelength. A 380-μm PDMS (Sylgard 184, Dow Corning Corp.) with a sound speed of 1140 m/s and a density of 960 kg/m$^3$ is adhered to the front of the composite material. A 100-μm stainless steel Fresnel spiral diffraction grating is placed above the PDMS, and the entire assembly is then cast in PDMS. An impedance analyzer (4294 A, Agilent) is utilized to characterize the impedance spectra of the KNN composite material, and a pulse-echo tester (5077 R, Agilent) is exploited to measure the pulse-echo characteristics of the device.

### Pressure field measurement of the f-FH

A customized scanning system is used to map the pressure field generated by the f-FH device. The device is connected to a function generator (AFG3252C, Tektronix) and a power amplifier (AG1020, T&C, USA), with 50 cycles of sine pulse set at different frequencies (2.7, 3.0 and 3.3 MHz). The pressure waves are captured by a hydrophone probe (Precision Acoustic, UK) placed in a deionized water tank. The hydrophone is positioned by a 3D precision moving stage (H2-2206, ESM, CN). The signals received by the hydrophone are recorded in the time domain using a data acquisition card (QT1140, Queentest, China) at a sampling rate of 125 MHz. Finally, these time-domain signals are Fourier-transformed in the processing system to extract the amplitude and phase values of the pressure field.

### Animal surgery

Sprague-Dawley male rats weighed 300 to 350 g are used in this study. All procedures are approved by the Animal Ethics Committee of Wuhan University Renmin Hospital. During the surgery and electrophysiological recording, rats are initially anesthetized with 5% isoflurane in oxygen and then placed on a stereotactic apparatus maintained with 1.5% to 2% isoflurane. Feedback heating pads are employed to maintain the rats' stable body temperature during the experiments.

### Construction of MI model

After anesthesia, mechanical ventilation is employed to maintain oxygen supply. A thoracotomy is performed in the third or fourth intercostal space on the left side, followed by ligating the left anterior descending coronary artery (LAD) with 6-0 silk suture to induce MI. MI occurrence is confirmed by ST-segment elevation or T-wave changes on electrocardiogram. The chest cavity is carefully closed, and the chest wall is sutured.

### Long-term stimulation on the PVN

The f-FH device is fixed on the skull above the PVN. Dental cement (Glass Ionomer Cement, Shangchi Dental Materials Co., Ltd., Changshu, China) is then applied. For the treatment of chronic MI, the rats implanted with the f-FH recover for 2 weeks, followed by MI modeling, and then underwent 15 min stimulation on the PVN daily for 4 weeks. The specific stimulation strategy and ultrasound parameter were detailed in Supplementary Table 2.

### Recording of LSG neural activity

LSG neural discharges are detected using a pair of platinum-coated electrodes and recorded by the PowerLab data acquisition system (8/35, AD Instruments, New South Wales, Australia). LSG neural activity, characterized by the frequency and amplitude of neural discharges, is defined as deviations with a signal-to-noise ratio greater than 3:1, consistent with our previous studies[69].

### Heart rate variability (HRV) analysis

HRV is measured to assess autonomic tone. Specifically, electrocardiogram recording is firstly performed by the PowerLab system. A 5-minute stable-state electrocardiogram is then analyzed using Lab-Chart 8.0 software. Finally, high-frequency power (HF, 0.75–2.5 Hz, reflecting parasympathetic tone), low-frequency power (LF, 0.25–0.75 Hz, reflecting sympathetic tone), and the ratio of LF to HF power (LF/HF) are processed[70].

### Detection of serum NE and NPY

Blood samples of rats are rapidly collected and centrifugated to obtain serum. The levels of serum NE and NPY are detected by enzyme-linked immunosorbent assay (ELISA) using NE ELISA kit (ELK7885, ELK Biotechnology, China) and NPY ELISA kit (ELK2448, ELK Biotechnology, China) according to the manufacturer's instructions.

### Assessment of infarct size

TTC staining is conducted to measure the myocardial infarct size. After finishing the experiment, the hearts of rats are rapidly extracted and frozen at −80 °C. The hearts are then sliced into 5-mm sections, immersed in 1.0% TTC solution, and incubated in the dark at 37 °C for 15 min. After rinsing with PBS, the slices are fixed in 4% paraformaldehyde solution overnight. Images are acquired by a digital camera and processed using Image-Pro Plus 6.0 software (MediaCybernetics, Carlsbad, CA, USA).

### Transmission electron microscope (TEM) of myocardium

The peri-infarct region of the heart is dissected into 1 mm$^3$ tissue blocks and immersed in an electron microscopy fixative solution (G1102, Servicebio, China). Subsequently, samples are fixed at room temperature in 1% osmium tetroxide for 2 h in a light-protected environment and subjected to gradient dehydration. Embedding is then performed using acetone and embedding agent (90529-77-4, SPI, USA) before sectioning. Finally, images are obtained by a transmission electron microscope (HT7800, HITACHI, Japan).

### Programmed electrophysiological stimulation (PES)

To assess the susceptibility to ventricular arrhythmias (VA), programmed electrical stimulation (PES) is applied to the left ventricle of rats. PES is set to deliver multiple consecutive pacing stimuli (S1) with a cycle length of 140 ms, followed by additional stimuli (S2, S3 and S4) at shorter intervals. The occurrence of VA is finally observed.

### Echocardiography

Echocardiography is carried out four weeks after MI to assess cardiac function utilizing a cardiac ultrasound diagnostic system (VIVID E95, GE, USA). Rats were anesthetized with 1.5% isoflurane and underwent B-mode and M-mode echocardiography beside the sternum. The following parameters are measured: left ventricular ejection fraction (EF%), fractional shortening (FS%), end-systolic volume (ESV), end-diastolic volume (EDV), end-systolic diameter (IDs), and end-diastolic diameter (IDd). All data are acquired by an experienced ultrasound physician who is blinded to the group information.

### Histological staining

After euthanasia, the brains, hearts, lungs, livers, spleens, and kidneys of rats are extracted and fixed in 4% paraformaldehyde, followed by embedding in paraffin and slicing into 5 μm. Immunofluorescence staining is performed using primary antibodies of anti-TH (GB11181, Servicebio, China), anti-Iba-1 (GB113502, Servicebio, China), anti-GFAP (GB11096, Servicebio, China), anti-P2X7R (28207-1-AP, Proteintech, China), anti-vWF (GB11020, Servicebio, China) and anti-α-SMA (GB12045, Servicebio, China) antibodies. Additionally, cell nuclei are labeled with 4,6-diamidino-2-phenylindole (DAPI). Masson's staining is used to evaluate myocardial interstitial fibrosis. Hematoxylin and eosin

(HE) staining and terminal deoxynucleotidyl transferase-mediated dUTP nick end labeling (TUNEL) staining are performed for the safety assessment of f-FH intervention.

## Biocompatibility studies

The temperature of the tissue surrounding the PVN is measured by a thermal imaging camera (FLIR C2, USA) to assess local temperature changes induced by f-FH operation. The scalp tissue above the PVN is extracted for HE staining to examine the effects of the device on scalp growth in rats. Additionally, blood samples are collected via right ventricular puncture for hematological and hepatic-renal function tests.

## Statistical analysis

In biological experiments, data are presented as Mean ± SEM. Statistical analysis is conducted using the $t$-test, or one-way ANOVA, with statistical significance set at $p < 0.05$. GraphPad Prism 10.0 is used for statistical analysis.

## Data availability

All data supporting the findings of this study are available within the article and its supplementary files. Any additional requests for information can be directed to, and will be fulfilled by, the corresponding authors. Source data are provided with this paper.

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

## Acknowledgements

This work was supported by the Natural Science Foundation of China under grant numbers U22A20259 (B.Z.) and 12304512 (T.Z.), the Shenzhen Basic Science Research under grant number JCYJ20200109110006136 (B.Z.), the Interdisciplinary Innovative Talents Foundation from Renmin Hospital of Wuhan University under grant number JCRCYG-2022-001 (S.W.), the Fundamental Research Funds for the Central Universities under grant number 2042023kf0182 (S.W.), the Natural Science Foundation of Hubei Province under grant numbers 2025AFB124 (T.Z.) and JCZRYB202400863 (S.W.) and Huazhong University of Science and Technology Research under grant number 2023JCYJ043 (B.Z.).

## Author contributions

B.Z. and S.W. conceived and designed experiments. T.Z., J.L., J.Z., D.K., H.H., M.T., J.O.-Y., X.Y., and B.Z. conducted the material, device preparation, and testing. T.Z., H.Y.H., Z.W., H.J., Y.C., W.G., and S.W. performed animal experiments, data collection, and analysis. T.Z., H.Y.H., S.W., and B.Z. wrote the manuscript. All authors discussed and commented on the manuscript.

## Competing interests

The authors declare no competing interests.
