## [Transparent Peer Review file · Nature Communications]

KNN-based frequency-adjustable ferroelectric heterojunction and biomedical applications

Corresponding Author: Professor Benpeng Zhu

Version 0:

Reviewer comments:

Reviewer #1

(Remarks to the Author)

The manuscript entitles "(K, Na) NbO₃-based lead-free ceramics with ultrahigh piezoelectricity for frequency-adjustable 1 ferroelectric heterojunction" is reporting high d₃₃ coefficient in KNN piezoceramics. KNN was then integrated in frequency adjustable ferroelectric heterojunction achieving focus adjustment in mm range in a frequency range 2.7-3.3 MHz. The device was then implanted and tested on rats for the transcranial nerve regulation and its effect on myocardial infarction. Individuals with implanted device showed slightly better results.

The reviewer is not expert for assessing experimental results after f-FH implantation in rats. This assessment is focused solely on the piezoelectric performance, as well as the manufacturing of the f-FH device and acoustic testing.

The reviewer believes that results of the manuscript are worth sharing with the scientific community.

The reviewer has following question for the authors:

1. Can you explain more how did you design the chemical composition of the KNN-3.5BHT?
2. How did you measure the d₃₃ coefficient ?
3. Can you explain what are the higher local relaxation characteristic shown in the Raman spectroscopy 1b?
4. Sentences in lines 21 to 23 are to be corrected, there is a dot "." instead of a comma ",", twice.

Thank you

Reviewer #2

(Remarks to the Author)

This is an interesting paper on the application of lead-free piezoelectric materials to frequency-tunable ferroelectric heterojunctions.

The paper claims improved performance due to improved properties of lead-free piezoelectric materials, but lacks a comparison with common lead-based materials. In addition to the improved properties as a lead-free piezoelectric material, this point should be mentioned in the paper to convince the reader of the superiority of this material.

In addition, there is a general lack of explanation. This is pointed out in detail below.

1. the improvement due to entropy needs to be compared with the d₃₃ value of pure KNN. The Raman scattering in Figure 1b should be explained in more detail. The method of measurement in Figure 1c should be noted. Please provide details and references for materials such as LF4 in Fig. 1d.
2. was AC polarization effective?
3. Is this material oriented? If oriented, what is the degree of orientation?
4. Figure 4E Impedance is usually written on a logarithmic axis.
5. fig. 4g Why does the frequency response swell to a frequency lower than 3Mhz?
6. figure 5f, how much does the acoustic impedance of the skull differ from the rest of the skull?

Reviewer #3

(Remarks to the Author)

In this work, Zhang et al. prepared KNN-based ceramics and a frequency-adjustable ferroelectric heterojunction (f-FH) was designed and prepared that achieved continuous focus adjustment in the millimetre range (1.5 mm) within a narrow frequency range (2.7-3.3 MHz). Although it is an interesting work and the data may be useful, this article fails on certain aspects.

A few concerns observed by the reviewer are provided below:

1. The authors claim that KNN-3.5BHT ceramics were prepared in the current study. Nevertheless, not a single piece of evidence is given to demonstrate that a single-phase perovskite structure was obtained. This study did not even adhere to presenting the XRD results, so the phase formation is an assumption without evidence.
2. How did the authors ensure that the composition maintains the exact atomic ratios:
(K_{0.505}Na_{0.5-95.5%})Ca_{0.01}Bi_{0.5-3.5%}(Nb_{0.965-95.5%}Sb_{0.035-95.5%}Zr_{0.01}Hf_{0.98-3.5%}Ti_{0.02-3.5%})O₃
Given the use of volatile oxides/carbonates such as K₂CO₃, Na₂CO₃ and Bi₂O₃, how was the nominal composition maintained? The authors need to provide evidence.
3. On both occasions, the composition is incorrectly written. The authors are advised to check it thoroughly.
4. On page 6, line no. 13, the authors mention "After poled at 30 kV cm⁻¹ by AC electric field at room temperature, the....."
Did the authors use DC or AC electric field for poling? Poling is done in DC electric field!!
5. The analysis section needs deeper interpretation of findings.
6. More discussion on the implications of the results in relation to existing work should be done.
7. The conclusion should summarize key findings more concisely and highlight their broader implications.
8. What is "NPB"?
9. The manuscript requires thorough English correction to improve readability and clarity. Issues such as grammatical errors, awkward phrasing, and inconsistencies in writing style should be addressed.

Overall, the paper addresses a pertinent topic within its field, however, given the above issues, I do not recommend publication of this manuscript in its present form.

Version 1:

Reviewer comments:

Reviewer #1

(Remarks to the Author)

The reviewer is satisfied with implemented modifications and has no more comments for the authors.
The manuscript can be published as it is.

Reviewer #2

(Remarks to the Author)

Thank you for your revisions and response.

The reviewers have read the response and the revised manuscript and find it to be appropriately revised and suitable for publication.

I think some of the AC polarization papers should be added to the bibliography.

Reviewer #3

(Remarks to the Author)

The authors provided satisfactory response to the comments raised.

We are very grateful to the editors and reviewers for their constructive comments. We have made significant efforts to conduct a series of analyses and explanations, and then revised the manuscript based on the reviewers' comments (Research Article, NCOMMS-25-16327A).

Our responses to the reviewer comments are shown below in BLUE and the reviewer comments in BLACK. Revised or added texts in the manuscript are printed in RED. Fig. x refers to the figures in the main manuscript, and Fig. Rx in this response.

Reviewer #1 (Remarks to the Author):

The manuscript entitles “(K, Na) NbO₃-based lead-free ceramics with ultrahigh piezoelectricity for frequency-adjustable ferroelectric heterojunction” is reporting high d₃₃ coefficient in KNN piezoceramics. KNN was then integrated in frequency adjustable ferroelectric heterojunction achieving focus adjustment in mm range in a frequency range 2.7-3.3 MHz. The device was then implanted and tested on rats for the transcranial nerve regulation and its effect on myocardial infarction. Individuals with implanted device showed slightly better results.

The reviewer is not expert for assessing experimental results after f-FH implantation in rats. This assessment is focused solely on the piezoelectric performance, as well as the manufacturing of the f-FH device and acoustic testing. The reviewer believes that results of the manuscript are worth sharing with the scientific community.

Response: We sincerely thank the reviewer for the important comments.

The reviewer has following question for the authors:

1. Can you explain more how did you design the chemical composition of the KNN-3.5BHT?

Response: Thanks for the question. This study aims to develop frequency-adjustable heterojunctions for long-term and high-precision transcranial neuromodulation. In order to achieve both environmental and performance demands, we adopt lead-free, high-performance KNN ceramics to ensure environmental sustainability and robust acoustic pressure output. As the entropy enhancement strategy, where multiple elements occupy equivalent lattice sites, enables the development of high-performance piezoelectric ceramics^[1,2], we design the composition of KNN-3.5BHT following a medium-entropy approach. Additionally, the corresponding descriptions have been incorporated into the manuscript, as shown in Fig. R1.

- [1] Yang, B. B. et al. High-entropy enhanced capacitive energy storage. Nat. Mater. 21, 1074 (2022).
- [2] Yuan, Q. et al. Achieving excellent energy storage performances in Bi_{0.5}Na_{0.5}TiO₃-based ceramics via a configuration entropy enhancement strategy. J. Alloys Compd. 1014, 178611 (2025).

62 Inspired by the morphotropic phase boundary in PZT ceramics, the construction of polymorphic phase
63 boundary by chemical modification can significantly improve the piezoelectricity³⁴. Moreover, the
64 entropy increase strategy^{35,36}, in which multiple elements occupying equivalent lattice positions has
65 recently provided key perspectives on the design of high-performance piezoelectric ceramics.

110 starting from the orthorhombic perovskite-type K_{0.5}Na_{0.5}NbO₃ matrix, medium-entropy piezoelectric
111 ceramics (Fig. 3a) are developed by introducing Sb, Zr, Hf, and Ti at the B-site, and Ca and Bi at the A-
112 site (Supplementary Fig. 1a), combined with the application of templated grain growth technology. The
113 resulting composition of the as-prepared material is (K_{0.505}Na_{0.5-95.5%}Ca_{0.01}Bi_{0.5-3.5%})
114 (Nb_{0.965-95.5%}Sb_{0.035-95.5%}Zr_{0.01}Hf_{0.98-3.5%}Ti_{0.02-3.5%})O₃ (abbreviated as KNN-3.5BHT). Crucially, the KNN-
115 3.5BHT ceramics demonstrate excellent compositional homogeneity (Supplementary Fig. 1b-c).

Figure R1. Added corresponding illustrations in the manuscript.

2. How did you measure the d₃₃ coefficient?

Response: The piezoelectric coefficient (d_{33}) is measured at room temperature using a quasi-static d_{33} meter (ZJ-6A, Institute of Acoustics, China). The temperature fatigue resistance of piezoelectricity is evaluated employing the GDPT-900A Variable Temperature Piezoelectric d_{33} Measurement System (JKZC, China). We also added the related descriptions in the manuscript, as illustrated in Fig. R2.

266 h in air. After poled at 30 kV/cm by AC electric field at room temperature, the piezoelectric coefficient
 267 (d_{33}) is tested at room temperature using a quasi-static d_{33} meter (ZJ-6A, Institute of Acoustics, China).
 268 The temperature fatigue resistance of piezoelectricity is evaluated using the GDPT-900A Variable
 269 Temperature Piezoelectric d_{33} Measurement System (JKZC, China), and the elemental composition and
 270 distribution are investigated using an EPMA-8050G electron probe microanalyzer (EPMA) (JEOL, Japan).

Figure R2. Added corresponding illustrations in the manuscript.

3. Can you explain what are the higher local relaxation characteristic shown in the Raman spectroscopy 1b?

Response: Thanks for the question. In the Raman spectra, compared with pure KNN, the high-entropy KNN-3.5BHT exhibits reduced relative intensities of specific Raman scattering peaks (marked by arrows) (Fig. R3a), implying that entropy enhancement induces greater local structural disorder and relaxation effects. Furthermore, we also have supplemented relevant descriptions in the manuscript, as shown in Fig. R3b.

Fig. R3. (a) Raman shift of the piezoelectric ceramics (KNN and KNN-3.5BHT); (b) Added in the manuscript.

4. Sentences in lines 21 to 23 are to be corrected, there is a dot “.” instead of a comma “,” twice.

Response: Thanks. We have corrected in the manuscript (Fig. R4).

77 **Design and field simulation of the f-FH**
 78 To address the key limitations of conventional transcranial focused ultrasound devices in
 79 neuromodulation applications, we propose a frequency-adjustable ferroelectric heterojunction (f-FH) for
 80 transcranial neuromodulation, which achieves continuous adjustable-focus at different operating
 81 frequencies (Fig.1b). By integrating ferroelectric KNN with a Fresnel spiral diffraction grating (Fig.1c),
 82 the device achieves a compact, fully implantable form factor.

Fig. R4. Revised corresponding descriptions in the manuscript.

Reviewer #2 (Remarks to the Author):

This is an interesting paper on the application of lead-free piezoelectric materials to frequency-tunable ferroelectric heterojunctions. The paper claims improved performance due to improved properties of lead-free piezoelectric materials, but lacks a comparison with common lead-based materials. In addition to the improved properties as a lead-free piezoelectric material, this point should be mentioned in the paper to convince the reader of the superiority of this material. In addition, there is a general lack of explanation. This is pointed out in detail below.

Response: We appreciate the reviewer's comments and have addressed each concern point-by-point as below.

Compared to lead-containing materials, the performance of developed KNN ceramics is superior to the widely used PZT (Fig. R5a). We also added the related descriptions in the manuscript, as depicted in Fig. R5b.

Fig. R5. (a) Revised figure in the manuscript; **(b)** Added in the manuscript.

1. the improvement due to entropy needs to be compared with the d_{33} value of pure KNN. The Raman scattering in Figure 1b should be explained in more detail. The method of measurement in Figure 1c should be noted. Please provide details and references for materials such as LF4 in Fig. 1d.

Response: Thank you for your evaluation and suggestion.

The d_{33} value of pure KNN is 150 pC/N, which is substantially lower than that of KNN-3.5BHT ($d_{33} = 680$ pC/N), as demonstrated in Fig. R6a.

Additionally, we analyze the Raman scattering spectra. Compared with pure KNN, the relative intensity of scattering peaks (marked by arrows) decreases gradually in the high-entropy KNN-3.5BHT, suggesting that entropy enhancement induces greater local structural disorder and relaxation effects (Fig. R6b). The temperature fatigue resistance of piezoelectricity is evaluated utilizing the GDPT-900A Variable Temperature Piezoelectric d_{33} Measurement System (JKZC, China) (Fig. R6c).

Finally, we provide details and references for materials including LF4 (Fig. R6d).

Fig. R6. (a) Added in the manuscript; (b) Raman shift of the piezoelectric ceramics (KNN and KNN-3.5BHT) and the additional description in the manuscript. (c) Added in the manuscript; (d) Revised Fig. 3c in the manuscript.

2. was AC polarization effective?

Response: Thanks for the question. Yes, AC polarization was effective. In fact, AC poling is one of the most important and widely adopted polarization methods for piezoelectric materials. Moreover, through a triple mechanism involving dynamic domain switching, stress relaxation, and domain structure optimization, AC poling maximizes the contribution of electric domains to mechano-electric coupling, thereby significantly enhancing the d_{33} [3-6].

[3] Sun, Y.et al. Recent progress on AC poling of relaxor-PbTiO₃ ferroelectric single crystals: a review. Jpn. J. Appl. Phys., 61, SB0802 (2022).

[4] Luo, C.et al. High temperature and low voltage AC poling for 0.24Pb(In₁/2Nb₁/2)O₃ 0.46Pb(Mg₁/3Nb₂/3)O₃-0.30PbTiO₃ piezoelectric single crystals manufactured by continuous-feeding Bridgman method. J.Materiomics, 7(3), 621-628 (2021).

[5] Kumar, A.et al. AC poling-induced giant piezoelectricity and high mechanical quality factor in [001] PMN-PZT hard single crystals. Sensors Actuat. A-Phys., 372, 115342 (2024).

[6] Ding, X.et al. Improving piezoelectric performance in PMN-0.26 PT single crystal via low frequency AC poling. Solid State Commun., 394, 115714 (2024).

3. Is this material oriented? If oriented, what is the degree of orientation?

Response: Yes, this material is oriented, and was prepared by the templated grain growth technology. As a result, it exhibits a preferred $\langle 001 \rangle$ orientation, which is confirmed by XRD (Fig. R7).

Fig. R7. (a) Added in the manuscript; (b) Revised of **Fig. 3b** in the manuscript.

4. Figure 4E Impedance is usually written on a logarithmic axis.

Response: Thanks for the suggestion. We have revised in the manuscript as displayed in **Fig. R8**.

Fig. R8. Revised **Fig. 3g** in the manuscript.

5. fig. 4g Why does the frequency response swell to a frequency lower than 3Mhz?

Response: Thanks for the question. By cutting and filling the KNN piezoelectric ceramics, a KNN-based composite material was obtained (**Fig. R9a**). The center frequency of this f-FH based on KNN composite is 3 MHz, and the -6 dB bandwidth reaches 60.5% (**Fig. R9b**). Therefore, its effective frequency can be lower than 3 MHz, laying the foundation for achieving a frequency-adjustable heterostructure.

Fig. R9. (a) Preparation of KNN piezoelectric composites in the manuscript; **(b)** The echo signal and frequency gain of the f-FH in the manuscript.

6. figure 5f, how much does the acoustic impedance of the skull differ from the rest of the skull?

Response: Thanks for the question. The acoustic impedance of the rat skull and brain tissue is approximately 5.3 MRayl and 1.6 MRayl, respectively, which leads to ultrasound attenuation. Nevertheless, our tests reveal an approximately linear relationship between the excitation voltage and transcranial focal pressure (**Fig. R10**), with an acoustic pressure of 0.36 MPa at 100 V, enabling deep brain neuromodulation.

(a) 146 within a narrow frequency band (2.7–3.3 MHz) (Supplementary Fig.6), and the focal width at 3.0 MHz
 147 reaches 480 μm at -6 dB (Fig.3o). Furthermore, the excitation voltage and transcranial focal pressure show
 148 an approximately linear relationship (Supplementary Fig.7), with an acoustic pressure of 0.36 MPa at 100
 149 V, enabling applications in deep brain neuromodulation.

Fig. R10. (a) Added in the manuscript; **(b)** Revised in the supplementary information.

Reviewer #3 (Remarks to the Author):

In this work, Zhang et al. prepared KNN-based ceramics and a frequency-adjustable ferroelectric heterojunction (f-FH) was designed and prepared that achieved continuous focus adjustment in the millimetre range (1.5 mm) within a narrow frequency range (2.7-3.3 MHz). Although it is an interesting work and the data may be useful, this article fails on certain aspects.

Response: We thank the reviewer for the constructive comments. We have addressed the reviewer's concerns point-by-point as below.

A few concerns observed by the reviewer are provided below:

1. The authors claim that KNN-3.5BHT ceramics were prepared in the current study. Nevertheless, not a single piece of evidence is given to demonstrate that a single-phase perovskite structure was obtained. This study did not even adhere to presenting the XRD results, so the phase formation is an assumption without evidence.

Response: We thank the reviewer for the comments and suggestions. We have supplemented the XRD experimental results, which confirm the perovskite structure of this ceramic material. As a result, it exhibits a preferred $\langle 001 \rangle$ orientation (Fig. R11).

Fig. R11. (a) Revised in the manuscript; (b) Revised Fig. 3b in the manuscript.

2. How did the authors ensure that the composition maintains the exact atomic ratios: $(K_{0.505}Na_{0.5 \cdot 95.5\%})Ca_{0.01}Bi_{0.5 \cdot 3.5\%}(Nb_{0.965 \cdot 95.5\%}Sb_{0.035 \cdot 95.5\%}Zr_{0.01}Hf_{0.98 \cdot 3.5\%}Ti_{0.02 \cdot 3.5\%})O_3$ Given the use of volatile oxides/carbonates such as K_2CO_3 , Na_2CO_3 and Bi_2O_3 , how was the nominal composition maintained? The authors need to provide evidence.

Response: Thanks for the question. During the preparation process, a certain amount of co-solvent and organic binder were added, and templated grain growth technology was employed, effectively ensuring the stability of the nominal composition (Fig. R12a.). Additionally, the prepared material was examined using Electron Probe Microanalyzer (EPMA), confirming that the KNN-3.5BHT ceramics exhibit excellent compositional uniformity (Fig. R12b-c.).

- (a) 261 ethyl alcohol and ZrO₂ balls after weighting. The ceramic slurry is prepared by mixing the dried calcined
 262 powder with ethanol/toluene co-solvents, organic binders, and 3 wt% high-quality NN templates. After 8
 263 hours of roller milling, the homogeneous slurry is casted using casting machine. The dried tapes are then
 264 cut, laminated, and pressed into pellets at 200–300 MPa and 60 °C for 10 min. Finally, all the pellets are
 265 undergone binder burnout at 600 °C (1 °C/min) followed by two-step sintering at 1180–1210 °C for 6–8
 266 h in air. After poled at 30 kV/cm by AC electric field at room temperature, the piezoelectric coefficient
- (b)
- (c) 113 resulting composition of the as-prepared material is (K_{0.505}Na_{0.5-95.5%}Ca_{0.01}Bi_{0.5-3.5%})
 114 (Nb_{0.965-95.5%}Sb_{0.035-95.5%}Zr_{0.01}Hf_{0.98-3.5%}Ti_{0.02-3.5%})O₃ (abbreviated as KNN-3.5BHT). Crucially, the KNN-
 115 3.5BHT ceramics demonstrate excellent compositional homogeneity (Supplementary Fig. 1b-c).

Fig. R12. (a) Revised in the manuscript; (b) Added in the supplementary information; (a) Added in the manuscript.

3. On both occasions, the composition is incorrectly written. The authors are advised to check it thoroughly.

Response: Thanks. We have made revisions in the manuscript, as shown in Fig. R13.

- 113 resulting composition of the as-prepared material is (K_{0.505}Na_{0.5-95.5%}Ca_{0.01}Bi_{0.5-3.5%})
 114 (Nb_{0.965-95.5%}Sb_{0.035-95.5%}Zr_{0.01}Hf_{0.98-3.5%}Ti_{0.02-3.5%})O₃ (abbreviated as KNN-3.5BHT). Crucially, the KNN-
 115 3.5BHT ceramics demonstrate excellent compositional homogeneity (Supplementary Fig. 1b-c).
- 255 **Preparation and test of the KNN**
- 256 The (K_{0.505}Na_{0.5(0.99-x%)}Ca_{0.01}Bi_{0.5-x%}) (Nb_{0.965(0.99-x%)}Sb_{0.035(0.99-x%)}Zr_{0.01}Hf_{0.98-x%}Ti_{0.02-x%})O₃ (x = 0,
 257 3.5, abbreviated as KNN-xBHT) piezoceramics are designed and fabricated using the conventional solid-

Fig. R13. Revised in the manuscript.

4. On page 6, line no. 13, the authors mention “After poled at 30 kV cm⁻¹ by AC electric field at room temperature, the.....” Did the authors use DC or AC electric field for poling? Poling is done in DC electric field!!

Response: AC polarization was effective in this work. In fact, AC poling is one of the most important and widely used polarization methods for piezoelectric materials. Moreover, through a triple mechanism involving dynamic domain switching, stress relaxation, and domain structure optimization, AC poling maximizes the contribution of electric domains to mechano-electric coupling, thereby significantly enhancing the d_{33} [3-6].

- [3] Sun, Y. et al. Recent progress on AC poling of relaxor-PbTiO₃ ferroelectric single crystals: a review. *Jpn. J. Appl. Phys.*, 61, SB0802 (2022).
- [4] Luo, C. et al. High temperature and low voltage AC poling for 0.24Pb(In_{1/2}Nb_{1/2})O₃ 0.46Pb(Mg_{1/3}Nb_{2/3})O₃-0.30PbTiO₃ piezoelectric single crystals manufactured by continuous-feeding Bridgman method. *J. Materiomics*, 7(3), 621-628 (2021).
- [5] Kumar, A. et al. AC poling-induced giant piezoelectricity and high mechanical quality factor in

[001] PMN-PZT hard single crystals. Sensors Actuat. A-Phys., 372, 115342 (2024).

[6] Ding, X. et al. Improving piezoelectric performance in PMN-0.26 PT single crystal via low frequency AC poling. Solid State Commun., 394, 115714 (2024).

5. The analysis section needs deeper interpretation of findings.

Response: We are grateful to the reviewer for the constructive suggestion. We conducted an in-depth analysis of the results:

Firstly, we analyzed the Raman scattering spectra. Compared with pure KNN, the relative intensity of scattering peaks (marked by arrows) decreases gradually in the high-entropy KNN-3.5BHT, implying that entropy enhancement induces greater local structural disorder and relaxation effects (Fig. R14a).

Additionally, through XRD and EPMA experiments, we thoroughly examined the perovskite structure and orientation of the KNN-3.5BHT material, as well as its excellent compositional uniformity (Fig. R14b).

Finally, we supplemented detailed animal experiments, including assessments of neuronal activity, neuroinflammatory responses, cardiac function, and biocompatibility. These confirmed that ultrasound stimulation of the PVN by the f-FH can effectively ameliorate MI (Fig. R14c).

Fig. R14. (a) Analysis of Raman spectroscopy; (b) XRD and EPMA experiments; (c) Supplemented detailed animal experiments.

6. More discussion on the implications of the results in relation to existing work should be done.

Response: We thank the reviewer for the valuable suggestion.

To address the limitations of traditional transcranial focused ultrasound devices in neuromodulation, we propose a frequency-adjustable ferroelectric heterojunction (f-FH). Given the

health concerns associated with lead-based piezoelectric materials and the requirement for effective acoustic pressure output, we developed high-performance lead-free KNN piezoelectric ceramics to fabricate the f-FH with high acoustic pressure output. The feasibility of the f-FH for MI treatment was further validated through animal experiments. Accordingly, we have revised the title to "KNN-based frequency-adjustable ferroelectric heterojunction and biomedical applications", updated relevant content, and added detailed validation of the animal experiment (Fig. R15).

36 abnormal neural activity. Unfortunately, traditional transcranial focused ultrasound devices encounter
37 substantial limitations in practical applications. Firstly, their large size and insufficient focusing precision
38 render them unsuitable for fullimplantation^{12,13}. Additionally, the fixed focal depth of existing devices¹⁴⁻
39 ¹⁶ constrains precise targeting of specific brain regions due to discrepancies between device design and
40 practical applications. Consequently, current transcranial focused ultrasound devices remain inadequate
41 for long-term, high-precision neuromodulation.

56 Additionally, piezoelectric materials are the core of ultrasonic devices, which is extremely important
57 for the development of ultrasonic heterojunction. Unfortunately, current piezoelectric devices mainly rely
58 on lead-based piezoelectric materials [e.g., Pb (Zr, Ti) O₃, PZT] as acoustic-electric coupling elements²⁹,
59 raising concerns over potential health hazards. In recent years, a growing number of studies have

106 As previously mentioned, the core component of the f-FH is a KNN-based ferroelectric ceramic. For
107 deep brain neuromodulation, KNN piezoceramics with high piezoelectricity are required to ensure
108 effective acoustic pressure output. Thus, based on the entropy phase transition strategy, ferroelectric KNN

Fig. R15. Revised in the manuscript.

7. The conclusion should summarize key findings more concisely and highlight their broader implications.

Response: According to the reviewer's comment, we have revised the conclusion (Fig. R16).

234 **Conclusion**

235 In summary, we design and fabricate a frequency-adjustable ferroelectric heterojunction (f-FH) based
236 on high-performance lead-free KNN-based piezoelectric ceramics (KNN-3.5BHT) with a d_{33} of 680 pC/N.
237 Acoustic field examination demonstrates the device's exceptional transcranial focusing capability,
238 achieving a transcranial focusing depth of 7.9 mm. The f-FH also enables continuous focal adjustment
239 over a 1.5 mm range under a narrow frequency range (2.7–3.3 MHz). *In vivo* implantation studies show
240 that the f-FH can achieve the therapeutic effect of MI after long-term high-precision ultrasound
241 neuromodulation of the PVN. Collectively, this study offers a new idea for the development and
242 application of KNN based ferroelectric heterojunctions and providing new insights into clinical treatments
243 for MI.

Fig. R16. Revised in the manuscript.

8. What is "NPB"?

Response: "NPB" is the abbreviation for "new phase boundary", and we have corrected in the manuscript (Fig. R17).

108 effective acoustic pressure output. Thus, based on the entropy phase transition strategy, ferroelectric KNN
109 ceramics with local polymorphic distortion of new phase boundary (NPB) are designed. Specifically,

Fig. R17. Revised in the manuscript.

9. The manuscript requires thorough English correction to improve readability and clarity. Issues such as grammatical errors, awkward phrasing, and inconsistencies in writing style should be addressed.

Response: We sincerely thank the reviewer for the valuable comment and suggestion. We have carefully revised the grammar and style of the manuscript.

We are very grateful to the editors and reviewers for their constructive comments. We revised the manuscript based on the reviewers' comments (Research Article, NCOMMS-25-16327B).

Reviewer #1 (Remarks to the Author):

The reviewer is satisfied with implemented modifications and has no more comments for the authors. The manuscript can be published as it is.

Response: We greatly appreciate the reviewer for your recommendation to publish our paper.

Reviewer #2 (Remarks to the Author):

Thank you for your revisions and response. The reviewers have read the response and the revised manuscript and find it to be appropriately revised and suitable for publication. I think some of the AC polarization papers should be added to the bibliography.

Response: Once again, we sincerely thank the reviewer for all the constructive suggestions and for the recommendation to publish our paper. We have added some AC polarization papers (Nat. Commun. [2024, 15, 2560], Solid State Commun. [2024, 15, 2560]) to the bibliography in the manuscript.

Reviewer #3 (Remarks to the Author):

The authors provided satisfactory response to the comments raised.

Response: Thank you for your feedback.